# Multifaceted role of galanin in brain excitability

**Nicolas N Rieser[1,2], Milena Ronchetti[1,2], Adriana Lea Lea Hotz[1], Stephan CF Neuhauss[1]\*†**

[1]Department of Molecular Life Sciences, University of Zurich, Zurich, Switzerland; [2]Life Science Zürich Graduate School, Zurich, Switzerland

**\*For correspondence:**
stephan.neuhauss@mls.uzh.ch

**Present address:** †Functional Genomics Center Zurich (FGCZ), University of Zurich, Switzerland

**Competing interest:** The authors declare that no competing interests exist.

## eLife Assessment

The study investigated the effects of the peptide galanin on brain Ca2+ activity in zebrafish, which provides a **useful** model organism for whole-brain imaging because of its transparency. They found that galanin has distinct effects on hyperactivity and expression of galanin changes after activity increases. The strength of evidence was **incomplete** particularly for some of the conclusions regarding the use of convulsants and relevance to epilepsy because of limitations to the methods and interpretations of results.

**Abstract** Galanin is a neuropeptide, which is critically involved in homeostatic processes like controlling arousal, sleep, and regulation of stress. This extensive range of functions aligns with implications of galanin in diverse pathologies, including anxiety disorders, depression, and epilepsy. Here, we investigated the regulatory function of galanin on whole-brain activity in larval zebrafish using wide-field $Ca^{2+}$ imaging. Combining this with genetic perturbations of galanin signaling and pharmacologically increasing neuronal activity, we are able to probe actions of galanin across the entire brain. Our findings demonstrate that under unperturbed conditions and during epileptic seizures, galanin exerts a sedative influence on the brain, primarily through the galanin receptor 1 a (*galr1a*). However, exposure to acute stressors like pentylenetetrazole (PTZ) compromises galanin's sedative effects, leading to overactivation of the brain and increased seizure occurrence. Interestingly, galanin's impact on seizures appears to be bidirectional, as it can both decrease seizure severity and increase seizure occurrence, potentially through different galanin receptor subtypes. This nuanced interplay between galanin and various physiological processes underscores its significance in modulating stress-related pathways and suggests its potential implications for neurological disorders such as epilepsy. Taken together, our data sheds light on a multifaceted role of galanin, where galanin regulates whole-brain activity but also shapes acute responses to stress.

## Introduction

Similar to classic neurotransmitters, neuropeptides are chemical messengers that are mainly synthesized and released by neurons. In contrast to conventional neurotransmitters, neuropeptides are amino acid chains between 3 and 36 amino acids in length. While classic neurotransmitters are primarily stored in synaptic vesicles, neuropeptides are predominantly stored in large dense-core vesicles (LDCV) and are mostly released during neuronal bursts or high-frequency firing (*Lang et al., 2015*; *van den Pol, 2012*). Neuropeptides play a crucial role in modulating neuronal activity and regulating various aspects of neural network function. They act as neuromodulators, exerting long-lasting effects on neuronal excitability, synaptic transmission, and plasticity (*Lang et al., 2015*; *Purves et al., 2019*). By influencing the balance between excitatory and inhibitory inputs to neurons, neuropeptides

can shape network dynamics and information processing. This is achieved through their interactions with specific receptors and signaling pathways, allowing for precise modulation and optimization of neural activity in response to changing conditions (*van den Pol, 2012*). Overall, neuropeptides play a pivotal role in orchestrating the complex activity of neuronal networks and maintaining homeostasis within the central nervous system. Elevated neuronal activity is often accompanied by shifts in neuropeptide transcription rates through a phenomenon known as 'stimulus-secretion-synthesis coupling' (*Douglas, 1968*; *MacArthur and Eiden, 1996*). This intricate interplay between neuropeptide release and biosynthesis likely serves as a crucial mechanism for replenishing neuropeptide stores, given the absence of a mechanism for neuropeptide uptake post-release and their degradation by extracellular peptidases. Furthermore, neuropeptides are well known to contact neurons via volume transmission, where extrasynaptically secreted molecules activate receptors on neurons synaptically unconnected to the releasing neuron. This leads to signal transmission across considerable distances, extending up to multiple microns, targeting multiple neurons in different regions of the brain (*Lang et al., 2015*; *van den Pol, 2012*; *Jan and Jan, 1982*; *Atkinson et al., 2021*; *Nässel, 2009*; *Ripoll-Sánchez et al., 2023*).

Galanin is such a neuropeptide, which is expressed predominantly in the hypothalamus, the center of homeostatic regulation in the brain. Galanin is highly involved in homeostatic processes like controlling arousal and sleep (*Woods et al., 2014*; *Podlasz et al., 2018*; *Reichert et al., 2019*; *Gaus et al., 2002*; *Kroeger et al., 2018*; *McGinty and Szymusiak, 2003*; *Ma et al., 2019*), but has been shown to regulate stress (*Corradi et al., 2022*; *Juhasz et al., 2014*; *Hökfelt et al., 2018*; *Khoshbouei et al., 2002*; *Picciotto et al., 2010*). Moreover, research has demonstrated that galanin-producing neurons in the hypothalamus play a role in regulating both food intake (*Schick et al., 1993*; *Adams et al., 2008*; *Qualls-Creekmore et al., 2017*; *Laque et al., 2015*; *Leibowitz et al., 1998*) and parental behavior (*Kohl et al., 2018*; *Wu et al., 2014*) in rodents. This extensive range of functions aligns with implications of galanin in diverse pathologies, including anxiety disorders, depression, and epilepsy (*Lang et al., 2015*; *Juhasz et al., 2014*; *Hökfelt et al., 2018*; *Kovac and Walker, 2013*; *Lerner et al., 2008*; *McColl et al., 2006*; *Fetissov et al., 2003*; *Jacoby et al., 2002*; *Mazarati et al., 2000*; *Drexel et al., 2018*). We found an upregulation of galanin in a recently described novel model of epilepsy (*Hotz et al., 2022*) that let us hypothesize that galanin may mediate a neuroprotective, net inhibitory effect on epileptic brains.

Here, we investigated the regulatory function of galanin on whole-brain activity in larval zebrafish (*Danio rerio*). Leveraging the transparency of zebrafish during their larval stages, a characteristic that facilitates live imaging, we utilized basic wide-field $Ca^{2+}$ imaging methods. Combining this with genetic perturbation of galanin and pharmacologically increasing neuronal activity, we were able to delve into the actions of galanin across the entire brain.

Our findings demonstrate that, under unperturbed conditions and during epileptic seizures, galanin exerts a net inhibitory influence on the brain, which is likely governed by *galanin receptor 1* a (*galr1a*). However, when faced with an acute stressor like pentylenetetrazole (PTZ), the typically sedative effects of galanin are substantially compromised. We found that exposure to this acute central nervous system stressor results in a galanin-dependent overactivation of the brain that overrides most of galanins sedating actions and increases the occurrence of epileptic seizures. Taken together, our data sheds light on a multifaceted role of galanin, where galanin regulates whole-brain activity but also shapes acute responses to stress.

## Results

### *gal* expression correlates with whole-brain activity

In prior work, we introduced a novel epilepsy model characterized by recurrent epileptic seizures and interictal neuronal hypoactivity (*Figure 1A-D*: *Hotz et al., 2022*). The unexpected observation of locomotor and neuronal hypoactivity in this model stands in contrast to most existing studies in zebrafish that report hyperactivity in epileptic animals (*Baraban et al., 2013*; *Hortopan et al., 2010*; *Baraban et al., 2005*). Intriguingly, recent investigations have demonstrated that overexpression of galanin induces locomotory hypoactivity in zebrafish models (*Woods et al., 2014*; *Podlasz et al., 2018*). To explore the potential involvement of galanin in the hypoactivity observed in *eaat2a* mutants, we conducted qPCR analysis on 5 dpf (days post fertilization) larval brains (*Figure 1C*). The results revealed

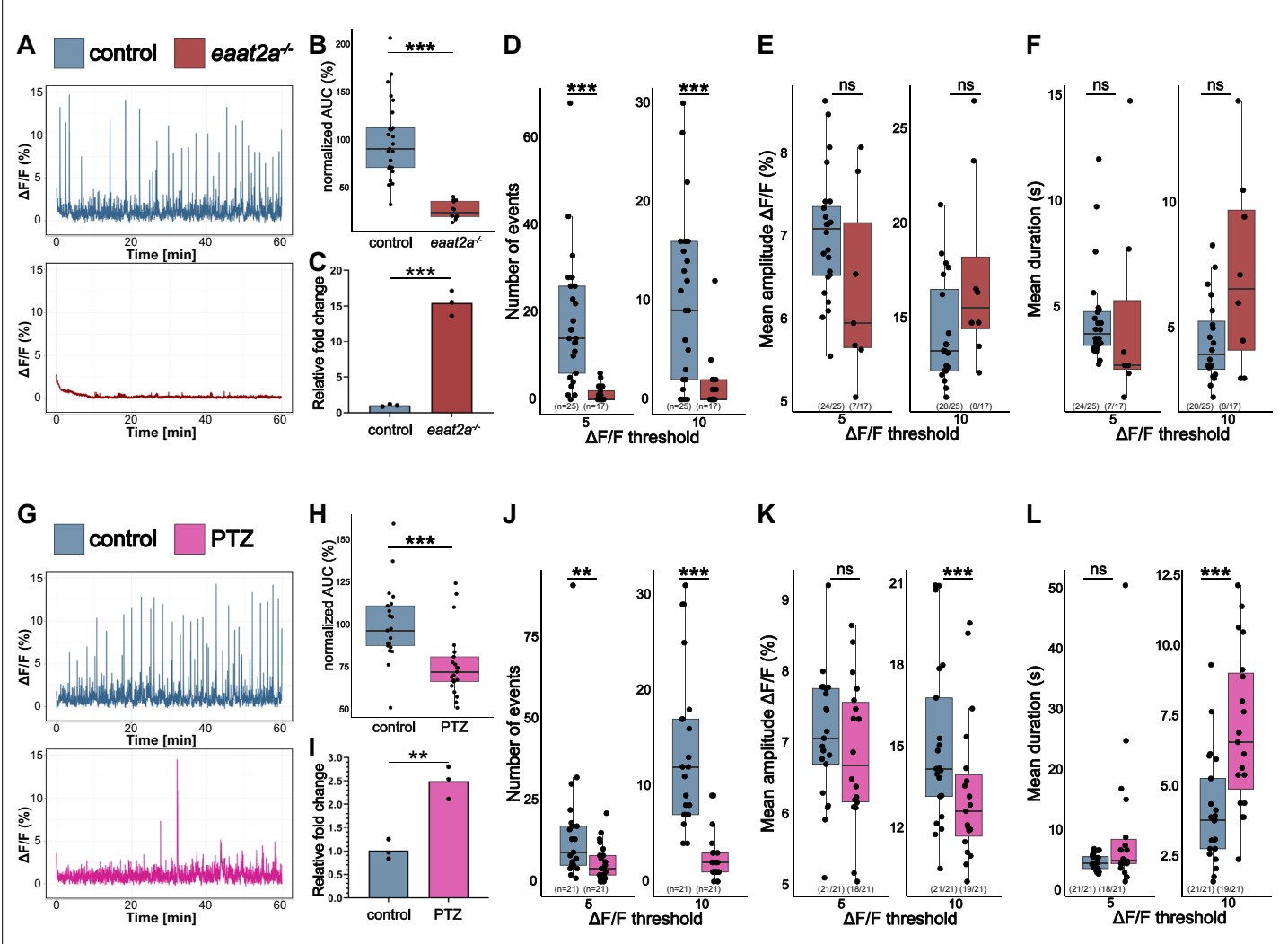

**Figure 1.** *gal* expression correlates with whole-brain activity. (**A**) Representative calcium signals (*elavl3:GCaMP5G*) recorded across the brain of 5 dpf control (*eaat2a+/+*) larva (blue, top) and *eaat2a-/-* mutant without seizure activity (red, bottom). (**B**) Area under the curve (AUC) calculated and averaged over two 5 min time windows per animal and normalized to control (n control = 25; n *eaat2a-/-* = 12). (**C**) Galanin transcript levels in pools of control vs. *eaat2a-/-* brains (n = 3). (**D**) Number of $Ca^{2+}$ events above 5% $\Delta F/F_0$ (left) and 10% $\Delta F/F_0$ (right). (**E**) Average amplitude of $Ca^{2+}$ events above 5% $\Delta F/F_0$ (left) and 10% $\Delta F/F_0$ (right) per larva. (**F**) Average duration of $Ca^{2+}$ events above 5% $\Delta F/F_0$ (left) and 10% $\Delta F/F_0$ (right) per larva. (**G**) Representative calcium signals (*elavl3:GCaMP6f*) recorded across the brain of 5 dpf control (blue, top) and larvae 2 hr after pentylenetetrazole (PTZ) exposure (PTZ rebound) (magenta, bottom). (**H**) AUC calculated over 1 hr recording, normalized to control (n control = 21; n PTZ rebound = 21). (**I**) Galanin transcript levels in pools of control vs. PTZ rebound brains (n = 3). (**G**) Number of $Ca^{2+}$ events above 5% $\Delta F/F_0$ (left) and 10% $\Delta F/F_0$ (right). (**J**) Average amplitude of $Ca^{2+}$ events above 5% $\Delta F/F_0$ (left) and 10% $\Delta F/F_0$ (right) per larva. (**K**) Average duration of $Ca^{2+}$ events above 5% $\Delta F/F_0$ (left) and 10% $\Delta F/F_0$ (right) per larva. Significance levels: \*\*\*p<0.001, \*\*p<0.01, \*p<0.05, ns = not significant (p>0.05), Wilcoxon-Mann-Whitney test (**B, E, H, J, K**), negative binomial regression (**D, I**), Student's t-test (**C, F**).

a significant 15.4-fold increase in galanin expression compared to wild-type siblings, suggesting an involvement of galanin in the interictal hypoactivity observed in *eaat2a* mutants. Furthermore, recent findings have shed light on the role of galanin in pharmacologically induced rebound sleep (*Reichert et al., 2019*). Additionally, it was demonstrated that increasing short-term neuronal activity results in subsequent inactivity that is dependent on galanin (*Reichert et al., 2019*). To explore whether the application of GABA_A receptor antagonist PTZ would also lead to a temporary decrease in whole-brain activity, we exposed 5 dpf larvae to 20 mM PTZ for 1 h, followed by a 2 hr washout period before undergoing $Ca^{2+}$ imaging. Remarkably, while PTZ increases swimming activity and induces seizure-like behavior during acute drug exposure (*Reichert et al., 2019*; *Baraban et al., 2005*; *Baraban et al., 2007*), there was a significant reduction in brain activity during PTZ rebound (*Figure 1G-K*), which

was correlated with an increase in galanin expression by 2.5-fold (*Figure 1I*) compared with non-PTZ-treated larvae. Notably, large $Ca^{2+}$ fluctuations ($\Delta F/F_0$ >10%) decreased in frequency (*Figure 1J*), and decreased in amplitude (*Figure 1K*), while their duration increased (*Figure 1L*) compared to non-PTZ-treated larvae. For small $Ca^{2+}$ fluctuations ($\Delta F/F_0$ >5) the frequency also decreased (*Figure 1J*), while their amplitude and duration were not affected (*Figure 1K and L*). Together, our results illustrate that the expression of galanin rises in reaction to seizure-like activity in two separate seizure models and is correlated with an overall decrease in whole-brain activity.

### *gal* controls whole-brain activity

To investigate if galanin by itself is able to reduce whole-brain activity, we employed a transgenic approach by using *hsp70l:gal* (*Woods et al., 2014*) expressing larvae. These transgenic fish express galanin under the control of the heat shock promoter *hsp70l*, enabling the induction of galanin expression by upregulating transcripts by over 300 fold (*Podlasz et al., 2018*). Attempts to induce galanin overexpression through heat shock were discontinued due to a notable impact on the brain activity of wild-type larvae compared to non-exposed wild-type larvae (*Figure 2—figure supplement 1*). Yet, considering that *hsp70l:gal* larvae already display a basal elevation of galanin transcripts by approximately eightfold (*Podlasz et al., 2018*), comparable with the upregulation seen in *eaat2a*⁻/⁻ mutants and PTZ rebound (*Figure 1C and I*), we chose to use larvae without inducing transcription via heat shock. We performed $Ca^{2+}$ imaging and found that indeed elevated expression of galanin in *hsp70l:gal* larvae leads to a decrease in whole-brain activity (*Figure 2A*) compared to wild-type siblings. We detected a reduction in frequency of $Ca^{2+}$ fluctuations for larger $Ca^{2+}$ events and a reduction in their amplitude (*Figure 2D and E*), mirroring our earlier findings in the PTZ rebound (*Figure 1G and H*). Notably, *hsp70l:gal* larvae exhibited no difference in small $Ca^{2+}$ fluctuations (*Figure 2D–F*). Subsequent quantification of galanin levels in the brain through qPCR unveiled a 1.8-fold increase in galanin transcripts (*Figure 2C*) in *hsp70l:gal* larvae compared to their wild-type siblings, potentially explaining the more subtle effects observed on brain activity. To further examine the role of galanin in regulating whole brain activity, we used *galanin*^t12ae/t12ae^ mutants (*gal*⁻/⁻ mutants) that carry a 7 bp deletion in exon 3 of the *galanin* gene which leads to a premature stop codon (*Eskova et al., 2020*). The absence of galanin protein was confirmed in an antibody staining showing a characteristic half-ring of galanin-positive fibers in the diencephalon that was completely absent in *gal*⁻/⁻ mutants (*Figure 2—figure supplement 2*). While other studies focused their analysis in *gal*⁻/⁻ mutants on their locomotion (*Reichert et al., 2019*; *Corradi et al., 2022*) and sleep (*Reichert et al., 2019*), we want to assess the effects of a galanin deficiency on the brain. When performing $Ca^{2+}$ imaging, we found that brain activity is heavily increased in *gal*⁻/⁻ mutants (*Figure 2G*) compared to wild-type larvae. While overexpression of galanin primarily influences larger $Ca^{2+}$ events, the absence of galanin specifically affects smaller $Ca^{2+}$ fluctuations. The increased frequency of these events is accompanied by a decrease in both amplitude and duration (*Figure 2I–K*). To delve deeper into the galanin system, we took a targeted approach focusing on a specific galanin receptor in zebrafish. Zebrafish have four galanin receptors: *galr1a*, *galr1b*, *galr2a*, and *galr2b*. While information about these receptors in zebrafish is limited, studies indicate that *galr1a* is widely expressed in the brain, and sequence identity closely resemble those of mammalian GALR1 (*Eskova et al., 2020*; *Li et al., 2013*; *Kim et al., 2014*). To investigate the actions of *galr1a* in zebrafish, we conducted F0 knockouts which enable a direct analysis of the injected embryos, bypassing the need to establish stable lines (*Kroll et al., 2021*). We found that *galr1a* crispants have an increased number of small $Ca^{2+}$ events in the brain, compared to control-injected larvae (*Figure 2O*). The absence of *galr1a* specifically affects the frequency of smaller $Ca^{2+}$ fluctuations, similar to *gal*⁻/⁻ mutants (*Figure 2I*), suggesting that galanin might regulate whole-brain activity via *galr1a*. We could not detect an upregulation of galanin in *galr1a* crispants, ruling out genetic compensation effects (*Figure 2N*).

Taken together, we could show that galanin is crucial in controlling spontaneous brain activity, and that this is at least partially regulated by *galr1a*.

### *gal* exerts only a modest effect on seizure activity in *eaat2a* mutants

While we could show that galanin has direct implications in regulating whole brain activity, others have investigated its antiepileptic properties. It was shown that galanin signaling can affect the seizure thresholds and severity in mice (*Fetissov et al., 2003*; *Jacoby et al., 2002*; *Mazarati et al., 2000*;

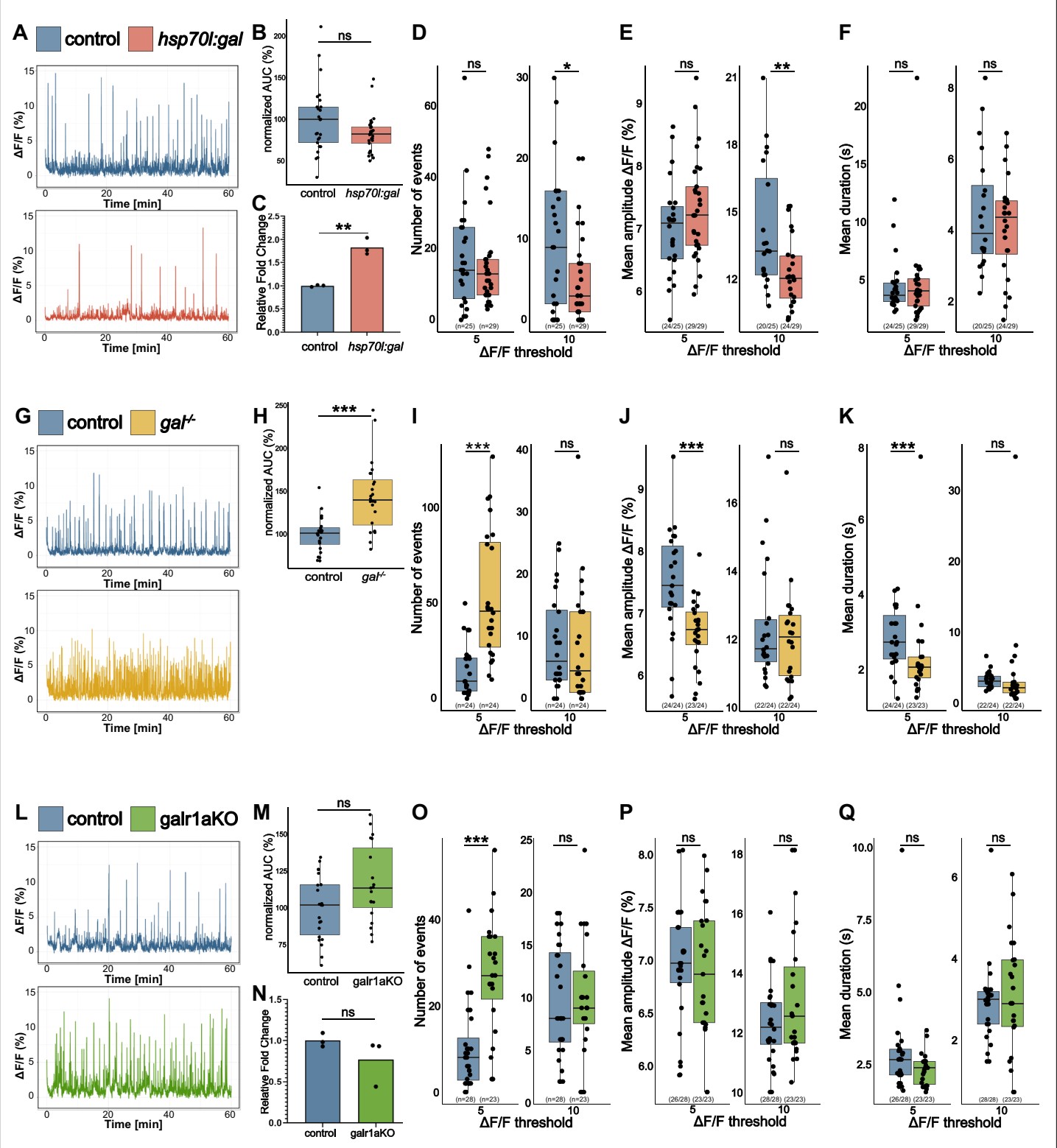

**Figure 2.** *gal* controls whole-brain activity. (**A**) Representative calcium signals (*elavl3:GCaMP5G*) recorded across the brain of 5 dpf control (blue, top) and *hsp70l:gal* sibling (orange, bottom). (**B**) Area under the curve (AUC) calculated over 1 hr recording, normalized to control (n control = 25; n *hsp70l:gal* = 29). (**C**) Galanin transcript levels in pools of control vs. *hsp70l:gal* brains (n = 3). (**D**) Number of Ca$^{2+}$ events above 5% $\Delta F/F_0$ (left) and 10% $\Delta F/F_0$ (right). (**E**) Average amplitude of Ca$^{2+}$ events above 5% $\Delta F/F_0$ (left) and 10% $\Delta F/F_0$ (right) per larva. (**F**) Average duration of Ca$^{2+}$ events above 5% $\Delta F/F_0$ (left) and 10% $\Delta F/F_0$ (right) per larva. (**G**) Representative calcium signals (*elavl3:GCaMP5G*) recorded across the brain of 5 dpf control (blue, top)

*Figure 2 continued on next page*

*Figure 2 continued*

and *gal*[-/-] larva (yellow, bottom). (**H**) AUC calculated over 1 hr recording, normalized to control (n control = 24; n *gal*[-/-] = 24). (**I**) Number of $Ca^{2+}$ events above 5% $\Delta F/F_0$ (left) and 10% $\Delta F/F_0$ (right). (**J**) Average amplitude of $Ca^{2+}$ events above 5% $\Delta F/F_0$ (left) and 10% $\Delta F/F_0$ (right) per larva. (**K**) Average duration of $Ca^{2+}$ events above 5% $\Delta F/F_0$ (left) and 10% $\Delta F/F_0$ (right) per larva. (**L**) Representative calcium signals (*elavl3:GCaMP6f*) recorded across the brain of 5 dpf control injected (blue, top) and galr1a crispants (galr1aKO) larva (green, bottom). (**M**) AUC calculated over 1 hr recording, normalized to control (n control = 28; n *galr1a* crispants = 23). (**N**) Galanin transcript levels in pools of control vs. galr1aKO brains (n = 3). (**O**) Number of $Ca^{2+}$ events above 5% $\Delta F/F_0$ (left) and 10% $\Delta F/F_0$ (right). (**P**) Average amplitude of $Ca^{2+}$ events above 5% $\Delta F/F_0$ (left) and 10% $\Delta F/F_0$ (right) per larva. (**Q**) Average duration of $Ca^{2+}$ events above 5% $\Delta F/F_0$ (left) and 10% $\Delta F/F_0$ (right) per larva. Significance levels: ***p<0.001, **p<0.01, *p<0.05, ns = not significant (p>0.05), Wilcoxon-Mann-Whitney test (**B, E, F, H, J, K, M, O, P**), negative binomial regression (**D**).

The online version of this article includes the following figure supplement(s) for figure 2:

**Figure supplement 1.** Heat shock decreases brain activity of 5 dpf wild-type larvae.

**Figure supplement 2.** Immunostaining confirms protein deficiency in *gal*[-/-] mutants whole-mount immunostaining of galanin localized in the diencephalon of 5 dpf zebrafish larvae.

*Drexel et al., 2018*), and can reduce the occurrence of seizure-like behavioral episodes and their intensity in zebrafish (*Podlasz et al., 2018*). The increased levels of galanin observed in *eaat2a*[-/-] mutants could be interpreted as an internal compensatory mechanism, possibly serving to shield neurons from excitotoxicity. The observed interictal hypoactivity may be a consequence of the brain's deliberate neuroprotective strategy so toduce the occurrence of seizures. To investigate if galanin could indeed act as a neuroprotective agent we, examined if additional galanin leads to seizure protection in *eaat2a*[-/-] mutants. *eaat2a*[-/-] mutants exhibit global spontaneous seizures lasting all the way up to over 6 min in duration (*Hotz et al., 2022*; *Figure 3*). Although not all *eaat2a*[-/-] mutants exhibit spontaneous epileptic seizures in the 1 hr recording window, an estimated 57.14–60.00% (12/21; 21/35) of larvae do (*Figure 3*). We then compared seizures of 5 dpf *eaat2a*[-/-];*hsp70l:gal* with *eaat2a*[-/-] larvae and found no significant difference in seizure susceptibility or severity in spontaneous epileptic seizures. Neither the number of seizures, their amplitude, duration nor time to peak was altered (*Figure 3A–F*). This is, however, not surprising, as we showed earlier that 5 dpf *eaat2a*[-/-] mutants already overexpress galanin by 15.4-fold (*Figure 1C*) and, therefore, more galanin might not make a big difference. To investigate if galanin is still important in seizure protection, we turned to analyze *eaat2a*[-/-];*gal*[-/-] mutants (*Figure 3G-L*). We found that area under the curve (AUC) of seizures is slightly increased (*Figure 3I*) and seizure amplitudes are increased compared to *eaat2a*[-/-] controls (*Figure 3K*). Neither the number of seizures, the seizure duration nor the time to peak changed significantly (*Figure 3H, J and L*). Taken together, galanin demonstrates only a modest level of seizure protection in *eaat2a*[-/-] mutants.

## *gal* promotes seizures in PTZ-exposed larvae

To investigate if galanin could mediate epileptic seizures in another state-of-the-art seizure model, we turned again to the $GABA_A$ receptor antagonist, PTZ. We applied 20 mM PTZ to the larvae directly before $Ca^{2+}$ imaging and compared PTZ-exposed larvae with their unexposed siblings. PTZ exposure increases neuronal activity in a progressive manner and triggers short epileptic seizures lasting up to 2 min in duration (*Hotz et al., 2022*; *Figure 4*).

Despite the observed decrease in neuronal activity resulting from galanin overexpression (*Figure 2A, D and E*), exposing 5 dpf *hsp70l:gal* larvae to 20 mM PTZ surprisingly led to a significant increase in seizure activity (*Figure 4D*) compared to non-exposed wild-type siblings. While the seizure number increases significantly (*Figure 4D*), there is also a decrease in seizure duration (*Figure 4F*). Conversely, the absence of galanin during acute PTZ exposure induces an opposing phenotype (*Figure 4J-K*). A marked decrease in the occurrence of epileptic seizures is evident in *gal*[-/-] mutants (*Figure 4K*) compared to wild-type larvae. Merely 6 out of 38 *gal*[-/-] mutants (15.8%) display seizures, whereas 20 out of 39 wild-type larvae (51.3%) exhibit seizure activity. However, when *gal*[-/-] larvae experience seizures upon PTZ, the seizure amplitude is significantly reduced (*Figure 4K*), while their duration (*Figure 4L*) increases. Furthermore, time to peak and AUC of seizures are increased (*Figure 4H–I*). Strikingly, while seizures in *eaat2a*[-/-] control larvae come to an abrupt stop after a gradual decline, seizures seem to taper off over the duration of several minutes in *gal*[-/-] mutants (*Figure 4G*).

Contrary to our initial expectations derived from prior results indicating a potentially sedative and neuroprotective role of galanin, the exposure of larvae to PTZ under acute conditions resulted in an opposing observation. While it was surprising that galanin seems to increase seizure susceptibility

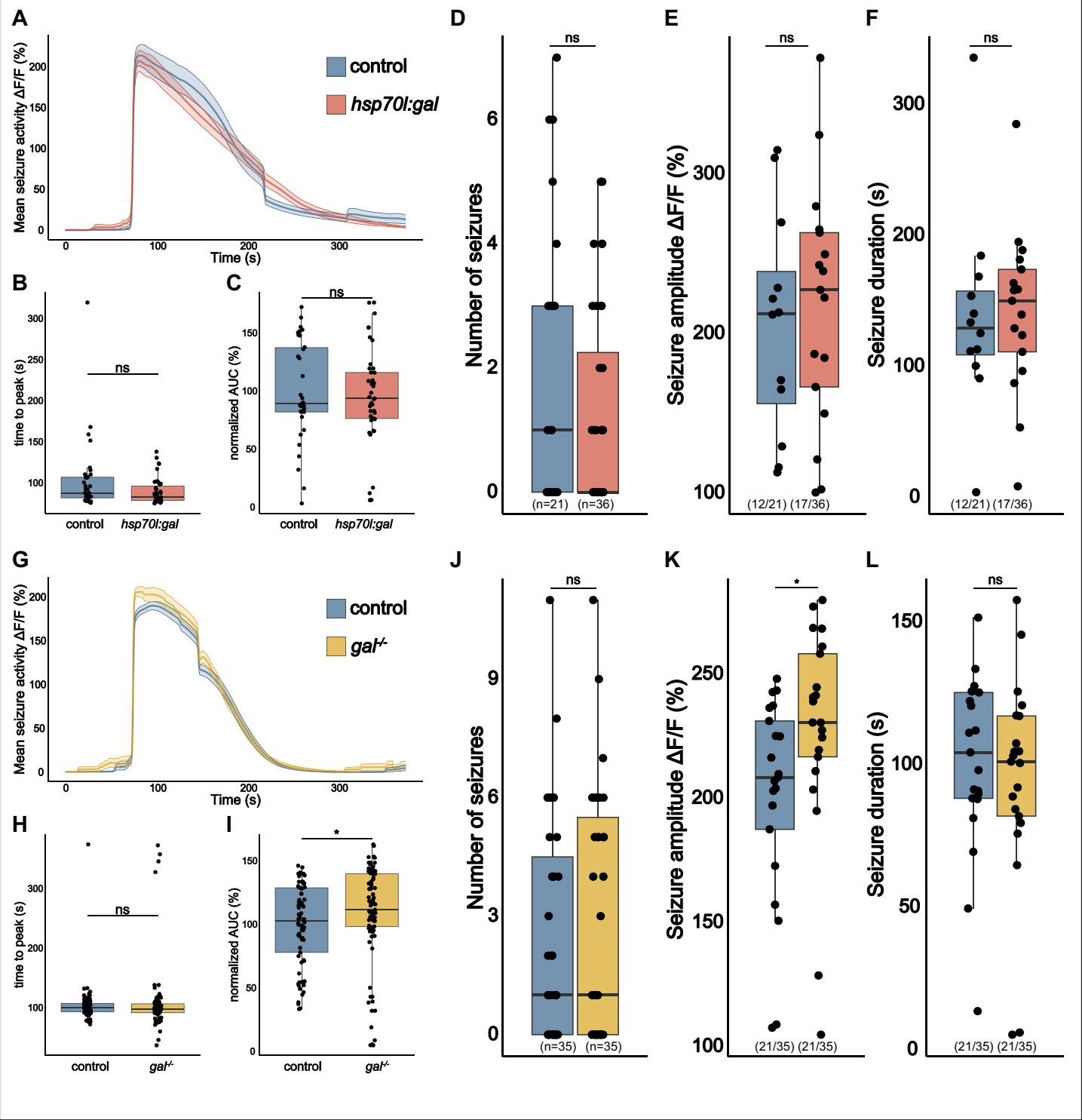

**Figure 3.** *gal* Exerts a modest effect on seizure activity in *eaat2a-/-* mutants. (**A**) Averaged calcium signals (*elavl3:GCaMP5G*) for spontaneous seizures recorded across the brain of 5 dpf control (*eaat2a-/-*) larva (blue, 35 events) and *eaat2a-/-;hsp70l:gal* (orange, 39 events) aligned by 50% of maximum amplitude. Shaded area represents SEM. (**B**) Time to peak calculated from beginning of aligned seizure until maximum ΔF/F$_0$ signal. (**C**) Area under the curve (AUC) calculated over spontaneous seizures normalized to control (n control = 35; n *eaat2a-/-;hsp70l:gal* = 39). (**D**) Number of spontaneous seizures per larva. (**E**) Amplitude of spontaneous seizures per larva. (**F**) Duration of spontaneous seizures per larva. (**G**) Averaged calcium signals (*elavl3:GCaMP5G*) for spontaneous seizures recorded across the brain of 5 dpf control (*eaat2a-/-*) larva (blue, 77 events) and *eaat2a-/-;gal-/-* (yellow, 89 events) aligned by 50% of maximum amplitude. Shaded area represents SEM. (**H**) Time to peak calculated from beginning of aligned seizure until maximum ΔF/F$_0$ signal. (**I**) AUC calculated over spontaneous seizures normalized to control (n control = 77; n *eaat2a-/-;gal-/-* = 89). (**J**) Number of

*Figure 3 continued on next page*

*Figure 3 continued*

spontaneous seizures per larva. (**K**) Amplitude of spontaneous seizures per larva. (**L**) Duration of spontaneous seizures per larva. Significance levels: ***p<0.001, **p<0.01, *p< 005, ns = not significant (p>0.05), Wilcoxon-Mann-Whitney test (**B, C, E, F, H, I, K, L**), negative binomial regression (**D, J**).

instead of protecting the larvae from seizures, this fits very well with results from a recent publication, where the authors found galanin to be involved in a stress response upon salient stimuli (*Corradi et al., 2022*). The authors hypothesize that galanin binds to autoreceptors on Gal+ neurons, reducing their own activity and, therefore, reducing their secretion of galanin and additionally reduce release of GABA on downstream stress-promoting neurons, ultimately leading to an increase in stress (*Corradi et al., 2022*). While the specific galanin receptor governing the stress pathway remains elusive, it has been found that GALR1 exerts an inhibitory influence on neuronal activity in mammals, while GALR2 displays a dual role, exhibiting both inhibitory and excitatory effects (*Lang et al., 2015*; *Ma et al., 2001*). Notably, it has been demonstrated that a significant portion of Gal+ neurons in the preoptic area in zebrafish express either *galr1a*, *galr1b*, or *galr2b* (*Corradi et al., 2022*). To discern whether *galr1a*, beyond its role in regulating whole-brain activity, is intricately linked to the stress response pathway, we implemented F0 knockouts of *galr1a* and exposed injected larvae to PTZ-induced stress. If *galr1a* were the autoreceptor on GABAergic Gal+ neurons, we would expect a decrease in seizure number similar to *gal*[-/-] mutants. However, our examination of *galr1a* crispants exposed to 20 mM PTZ revealed no significant decrease in seizure frequency compared to control injected larvae (*Figure 4P*). This suggests that *galr1a* is not the hypothesized autoreceptor governing this stress response pathway. Furthermore, our findings reveal that *galr1a* crispants manifest a more severe response to PTZ-induced seizures, evident in the substantial increase in both amplitude and duration compared to control injected larvae (*Figure 4Q and R*). Strikingly, several seizure characteristics in *galr1a* crispants mirror those observed in *gal*[-/-] mutants exposed to PTZ. Specifically, AUC of seizures shows a marked increase, the time to peak extends, and seizures exhibit a gradual tapering off over several minutes, a distinct pattern in contrast to the rapid resolution seen in controls (*Figure 4M–O*).

Taken together, our findings underscore the multifaceted role of galanin, not only in inducing a sedating effect on the entire brain but also in playing a pivotal role in a recently described stress response pathway (*Corradi et al., 2022*). Although the specific receptor orchestrating this stress pathway remains unidentified, our research decisively eliminates *galr1a* as the regulator. Collectively, this leads us to the conclusion that galanin likely exerts its sedative actions during epileptic seizures through *galr1a*, and that the stress response pathway is governed by another galanin receptor.

## Discussion

Sustaining brain homeostasis enables consistent behavior and demands the brain's ongoing integration of sensory information from the environment, internal behavioral state of the organism, and the elaborate networks of synaptic and non-synaptic connectivity (*Lin et al., 2022*). While maintenance of homeostasis requires multiple factors, here we focused on the neuropeptide galanin, as it has been shown to impact whole organism activity (*Woods et al., 2014*; *Podlasz et al., 2018*; *Reichert et al., 2019*, sleep *Reichert et al., 2019*; *Gaus et al., 2002*; *Kroeger et al., 2018*; *McGinty and Szymusiak, 2003*; *Ma et al., 2019*), and responses to stress (*Corradi et al., 2022*; *Khoshbouei et al., 2002*; *Picciotto et al., 2010*; *Tillage et al., 2021*; *Tillage et al., 2020*). In animals such as zebrafish, sleep is operationally defined as periods of locomotor quiescence with an increased arousal threshold (*Barlow and Rihel, 2017*; *Hendricks et al., 2000*; *Raizen et al., 2008*; *van Alphen et al., 2013*). A recent study showed evidence for galanin being important in homeostatic sleep regulation in zebrafish (*Reichert et al., 2019*). Others found a decrease in locomotor activity upon galanin overexpression (*Woods et al., 2014*; *Podlasz et al., 2018*). We similarly found that overexpression of galanin leads to a reduction in brain activity, while lack of galanin leads to neuronal hyperactivity. Interestingly, most sleep-active GABAergic neurons in the ventrolateral preoptic area (VLPO) are galanin positive in mammals and the VLPO has long been proposed to be a major sleep regulator (*Kroeger et al., 2018*; *Gaus et al., 2002*; *Lu et al., 2000*). While our investigation did not delve into galanin's direct influence on sleep, our results suggest that galanin exerts a sedative influence on the brain. This calming effect is likely governed by *galr1a*, as lack of *galr1a* leads to similar neuronal hyperactivity of the brain as the absence of galanin. This is in line with the observation that GALR1 activation inhibits the expressing

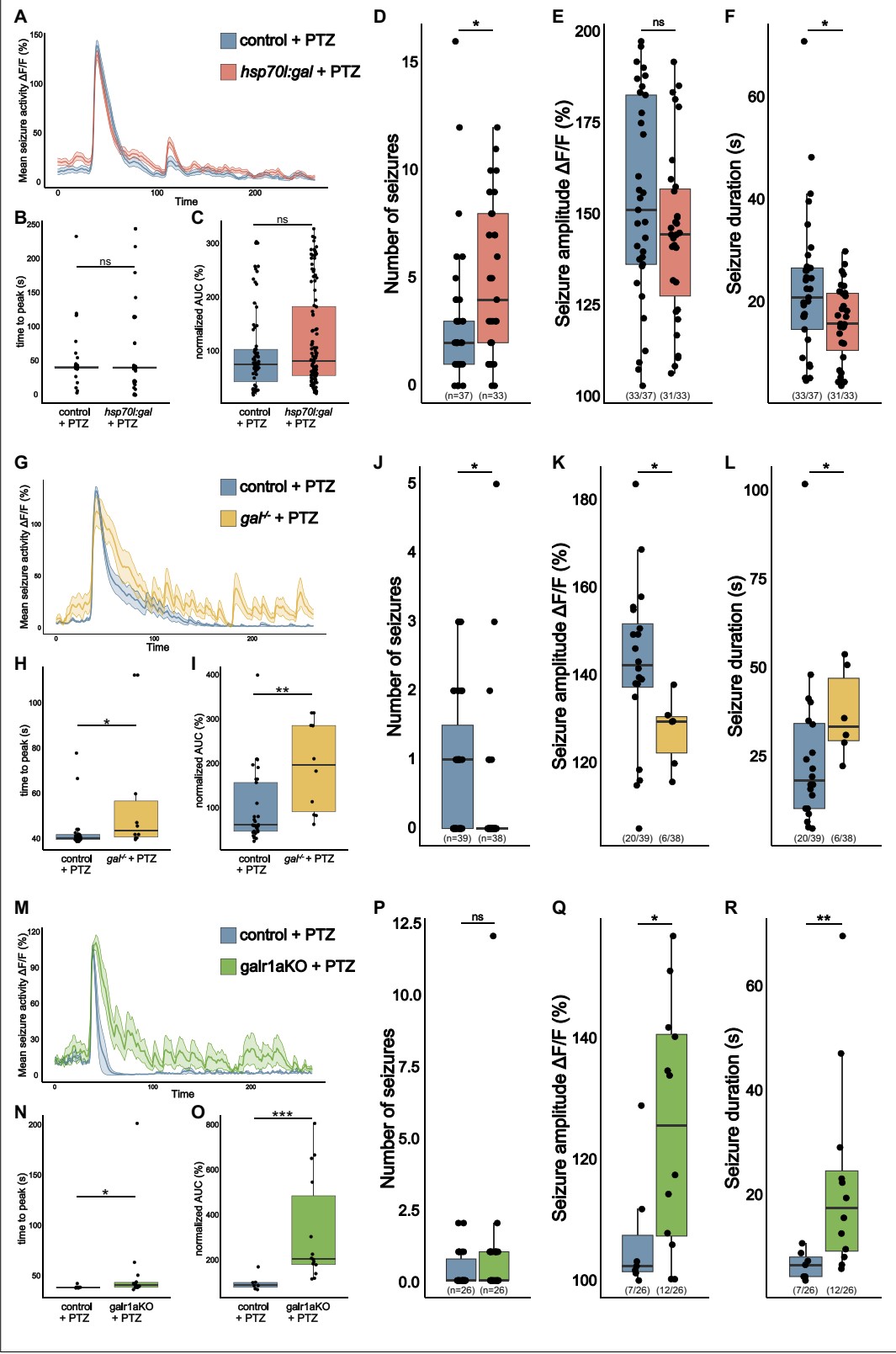

**Figure 4.** gal promotes seizures in pentylenetetrazole (PTZ) exposed larvae. (**A**) Averaged calcium signals (*elavl3:GCaMP6f*) for seizures elicited by 20 mM PTZ recorded across the brain of 5 dpf control (blue, 76 events) and *hsp70l:gal* (orange, 113 events) aligned by 50% of maximum amplitude. Shaded area represents SEM. (**B**) Time to peak calculated from beginning of aligned seizure until maximum $\Delta F/F_0$ signal. (**C**) Area under the

*Figure 4 continued on next page*

*Figure 4 continued*

curve (AUC) calculated over seizures normalized to control (n control = 76; n *hsp70l:gal* = 113). (**D**) Number of seizures per larva. (**E**) Amplitude of seizures per larva. (**F**) Duration of seizures per larva. (**G**) Averaged calcium signals (*elavl3:GCaMP6f*) for seizures elicited by 20 mM PTZ recorded across the brain of 5 dpf control (blue, 29 events) and *gal^-/-* (yellow, 10 events) aligned by 50% of maximum amplitude. Shaded area represents SEM. (**H**) Time to peak calculated from beginning of aligned seizure until maximum $\Delta F/F_0$ signal. (**I**) AUC calculated over seizures normalized to control (n control = 29; n *gal^-/-* = 10). (**J**) Number of seizures per larva. (**K**) Amplitude of seizures per larva. (**L**) Duration of seizures per larva. (**M**) Averaged calcium signals (*elavl3:GCaMP6f*) for seizures elicited by 20 mM PTZ recorded across the brain of 5 dpf control injected (blue, seven events) and *galr1a* crispants (galr1aKO, green, 14 events) aligned by 50% of maximum amplitude. Shaded area represents SEM. (**N**) Time to peak calculated from beginning of aligned seizure until maximum $\Delta F/F_0$ signal. (**O**) AUC calculated over seizures normalized to control (n control = 7; n *galr1a* crispants = 14). (**P**) Number of seizures per larva. (**Q**) Amplitude of seizures per larva. (**R**) Duration of seizures per larva. Significance levels: ***p<0.001, **p<0.01, *p<0.05, ns = not significant (p>0.05), Wilcoxon-MannWhitnMann Whitney (**B, C, E, F, H, I, K, L, N, O, Q, R**), negative binomial regression (**D, J, P**).

cell by opening G protein-regulated inwardly rectifying K^+ channels (GIRKs) (*Lang et al., 2015*). Moreover, it has been shown that galanin signaling is implicated in the reduction of glutamate release (*Kokaia et al., 2001*; *Elliott-Hunt et al., 2004*; *Zini et al., 1993*), which could further explain its calming effect on the brain.

Besides galanin regulating global neuronal activity, we showed that excessive neuronal activity is a potential predictor of subsequent reduced neuronal inactivity similar to another study (*Reichert et al., 2019*). Employing two distinct models characterized by increased neuronal activity and epileptic seizures, our findings elucidate a consistent pattern – elevated galanin levels and decline of overall neuronal activity. This discovery is particularly interesting, considering that post-ictal fatigue is a common symptom in epilepsy patients (*Hamelin et al., 2010*; *Ettinger et al., 1999*) and it further mimics slow background brain activity and reduced muscle tone present in human patients (*Allen et al., 2013*; *Epi4K Consortium, 2016*; *Guella et al., 2017*). Moreover, the reduced neuronal activity is particularly intriguing, especially when considering the divergent mechanisms of action utilized by the two models we employed. In *eaat2a^-/-* mutants glutamate uptake into astroglia is deprecated. This results in an accumulation of glutamate in the synaptic cleft, leading to the initiation of severe epileptic seizures that spread across the entire brain (*Hotz et al., 2022*). PTZ, on the other hand, acts as a GABA_A receptor antagonist, leading to the disinhibition of the brain and a subsequent dose-dependent increase in neuronal activity, ultimately triggering epileptic seizures. The observation that both models show a ruced neuronal activity interictally (*eaat2a^-/-*) or after washout (PTZ), which coincides with upregulation of galanin, suggests possible implications of galanin in various types of epileptic seizures. Galanin has been shown to be antiepileptic in numerous studies (*McColl et al., 2006*; *Fetissov et al., 2003*; *Jacoby et al., 2002*; *Mazarati et al., 2000*; *Kokaia et al., 2001*). It should also be noted that Galanin is known to be depleted by seizures in the hippocampus of mice (*Mazarati et al., 1998*). In an attempt to assess potential antiepileptic effects of galanin in zebrafish, we manipulated galanin levels in *eaat2a^-/-* mutants. However, the impact of galanin on seizures in *eaat2a^-/-* larvae was found to be minimal, likely attributed to the exceptionally severe seizure phenotype associated with *eaat2a* dysfunction (*Hotz et al., 2022*). In acute PTZ exposure, however, galanin modulated seizures significantly. While we hypothesized to see antiepileptic effects upon modulation of galanin, our observations did not initially support this expectation. Galanin overexpression significantly elevated the number of seizures, whereas the absence of galanin led to a decrease in seizure occurrence. This is, however, in line with results from a recent publication, where galanin was found to be involved in fine-tuning a stress response upon exposure to hypertonic solution (*Corradi et al., 2022*). Supposedly, galanin binds to autoreceptors on Gal^+ neurons, reducing their own activity and, therefore, reducing their release of GABA on downstream stress-promoting neurons, ultimately leading to an increase in stress via Crh^+ neurons (*Corradi et al., 2022*). This can be thought of as a control system to fine-tune neuroendocrine and behavioral responses to stressful situations. While it is not clear which galanin receptor is governing this inhibitory loop, *galr1a* stands out as the most expressed and main inhibitory galanin receptor in the zebrafish brain (*Eskova et al., 2020*; *Li et al., 2013*; *Kim et al., 2014*). Hence, we explored whether *galr1a* crispants exhibit a reduction in seizure occurrence similar to *gal^-/-* mutants upon PTZ-induced stress. Surprisingly, we found that seizures did

not decrease in number in *galr1a* crispants, indicating that *galr1a* might not function as the autoreceptor on GABAergic Gal⁺ neurons. Furthermore, seizures got more severe in *galr1a* crispants, as their amplitude and duration increased significantly, similar as in *gal⁻ᐟ⁻* mutants. This suggests that galanin influences whole brain activity in at least two ways during epileptic seizures. On one hand, galanin acts as a sedating agent, decreasing seizure severity, which is likely governed by *galr1a*. On the other hand, galanin increases seizure occurrence, similar to increasing stress in another study (*Corradi et al., 2022*) and we hypothesize that this might be dependent on one of the other galanin receptors.

Taken together, our data sheds light on a multifaceted role of galanin in regulating whole brain activity. The complexity of galanin's effects is evident in our study, where it can serve as a central nervous system depressant or regulate stress response pathways. This is particularly the case during acute stress, where its modulatory role overrides its sedative actions. This suggests a nuanced interplay between galanin and various physiological processes, elucidating on its potential significance in modulating stress-related pathways. Exploring the mechanisms underlying these effects could offer valuable insights into the intricate role of galanin and its implications for various physiological functions or neurological disorders such as epilepsy. Other studies have revealed that galanin-producing neurons in the hypothalamus are involved in regulating diverse processes including food intake (*Schick et al., 1993*; *Adams et al., 2008*; *Qualls-Creekmore et al., 2017*; *Laque et al., 2015*; *Leibowitz et al., 1998*), parental behavior (*Kohl et al., 2018*; *Wu et al., 2014*), sleep (*Reichert et al., 2019*; *Gaus et al., 2002*; *Kroeger et al., 2018*; *McGinty and Szymusiak, 2003*; *Ma et al., 2019*), and stress (*Corradi et al., 2022*; *Juhasz et al., 2014*; *Hökfelt et al., 2018*; *Khoshbouei et al., 2002*; *Picciotto et al., 2010*). Moreover, discrete subsets of hypothalamic neurons express galanin, with each subset displaying distinct responses to various stimuli. This diversity in neuronal responsiveness may account for the broad spectrum of physiological functions governed by galanin within the brain (*Corradi et al., 2022*). Investigating the function of these galaninergic neuron populations represents a crucial next step in unraveling how galanin modulates neuronal activity and ultimately influences brain function. Although many questions remain, our data reveals a multifaceted role of galanin, where galanin regulates whole-brain activity but also shapes acute responses to stress and epileptic seizures.

## Materials and methods

### Zebrafish lines and maintenance

All zebrafish experiments were performed on larvae at 5 d post fertilization (dpf). Zebrafish were kept under standard conditions at 28 °C on a 14 hr/10 hr light/dark cycle (*Mullins et al., 1994*). For experiments, adult animals were set up pairwise, and embryos and larvae were raised in E3 medium (5 mM NaCl, 0.17 mM KCl, 0.33 mM CaCl2, 0.33 mM MgSO4). All experiments were conducted in accordance with local authorities (Zürich, Kantonales Veterinäramt TV4206). The study's developmental stages preclude the determination of zebrafish sex. Animals were randomly allocated to experimental groups.

The following previously established transgenic lines were used in this study and combined according to needs:

> *Tg(elavl3:GCaMP5G)* **Akerboom et al., 2012**; *Tg(elavl3:GCaMP6f)* **Chen et al., 2013**; *Tg(hsp70l:gal)ᵃ¹³⁵* **Woods et al., 2014**; *eaat2aᶻʰ⁷* (*Hotz et al., 2022*); *galᵗ¹²ᵃᵉ ⁴¹*

Unless otherwise stated, experiments were performed with *nacre⁻ᐟ⁻* (Ca²⁺ imaging and qPCR) and AB strain (qPCR) (RRID:ZIRC_ZL1) zebrafish.

### Genotyping

Larvae were genotyped after each experiment for *hsp70l:gal*, *eaat2a*, and *galr1a*. For experiments with *galᵗ¹²ᵃᵉ/ᵗ¹²ᵃᵉ* mutants, adults were genotyped, kept in homozygosity, and subsequently incrossed for experiments. For genotyping, whole larvae were anesthetized on ice, lysed in base solution (25 mM NaOH, 0.2 mM EDTA), and neutralized in neutralization solution (40 mM Tris-HCl), before PCR was performed. The target site for *hsp70l:gal* (primers: 5'-CTTGTTGACTAGAAAAATCCTTTCA-3' and 5'-TCTCTCTTTCCTGCCAGTCC-3'), *eaat2a* (primers: 5'-GATGCAGTCGTATGGGAA-3' and 5'-CCTTCTCCCAGATTCTCC-3'), and *galr1a* (primers: 5'-ATGGGCACCCAAAACAACAGT-3' and 5'-GCGGCAGCACATATCCAAAAA-3') was PCR amplified with a fast-cycling polymerase (KAPA2G Fast

HotStart PCR kit, KAPA Biosystems) and a subsequent gel-electrophoresis, to allow detection of the mutant fragments.

## Quantitative real-time PCR (qPCR)

5 dpf larvae were anesthetized on ice and brains dissected using an insect pin and a syringe needle in a dish containing RNA later (Sigma-Aldrich). To confirm genotype, remaining tissue was lysed and gDNA amplified by means of PCR (KAPA Biosystems) as described above. Total RNA of equal pools of (n=15) larval brains was extracted using the RNeasy kit (Qiagen). RNA was reverse transcribed to cDNA using the Super Script III First-strand synthesis system (Invitrogen). The qPCR reactions were performed using SsoAdvanced Universal SYBR Green Supermix on a CFX96 Touch Real-Time PCR Detection System (Bio-Rad). *actb1*, *tbp* and *rpl13a* were chosen as reference genes. Primer efficiencies for *gal* (5'-TGAGGATGTCGTCCATACCATC-3' and 5'-GGTTGACTGATCTCTTCTGATGTG-3'), *actb1* (5'-CAGACATCAGGGAGTGATGGTTGG-3' and 5'-CAGATCTTCTCCATGTCATCCCAG-3'), *tbp* (5'-ACAACAGCCTACCTCCTTTCG-3' and 5'-CGTCCCATACGGCATCATAGG-3') and *rpl13a* (5'-TCTG GAGGACTGTAAGAGGTATGC-3' and 5'-AGACGCACAATCTTGAGAGCAG-3' *Lin et al., 2009*) were calculated by carrying out a dilution series and specificity was determined using melting curve analysis. After primer efficiencies were determined equal, brain samples were used for qPCR using 1 ng of cDNA per reaction. The controls 'no reverse transcription control' (nRT control) and 'no template control' (NTC) were performed with every qPCR reaction. All reactions were performed in technical triplicates. Data was analyzed in CFX Maestro Software from Bio-Rad and GraphPad Prism version 10.1.2 for Windows, GraphPad Software, Boston, Massachusetts USA, https://www.graphpad.com/. Expression of *gal* was normalized to expression of *actb1*, *tbp,* and *rpl13a*. Relative expression levels were calculated using the ΔΔCt method. Each experiment was performed in three biological replicates.

## F0 knockouts

We performed F0 knockouts similar as described in *Kroll et al., 2021*; *Kroll and Rihel, 2021*; *Kroll, 2022*. In brief, three CRISPR target sites for *galr1a* (*ENSDARG00000005522*) were selected with CHOPCHOP *Labun et al., 2019*. While it was not possible to generate efficient guides targeting 3 different exons, we chose to generate 3 guide RNAs #1 (5'-CGCCTACTATCAGGGCATCGTGG-3'), #2 (5'-TGGCGTCCTTGGAAACTCTCTGG-3'), #3 (5'-GGTGAAGATGCTTACGAGCATGG-3') targeting the same exon (exon 1). Reagents were ordered from integrated DNA technologies (IDT) and were resuspended, annealed, and pooled according to *Kroll and Rihel, 2021*. For generating the F0 knockouts, one-cell staged embryos were injected with 1 nl injection mix (10.1 fmol [357 pg] per gRNA) into the yolk. Control injection mix contained a set of three scrambled RNPs, which carry gRNAs IDT, Alt-R CRISPR-Cas9 Negative Control crRNA #1 (#1072544), #2 (#1072545), #3 (#1072546) (*Kroll et al., 2021*) with pseudo-random sequences predicted to not match any genomic locus. For all experiments, scramble RNP-injected siblings were used as controls. As stable *galr1a* mutants (*Eskova et al., 2020*) lack overt morphological phenotypes apart from their pigmentation phenotype, all larvae with abnormal appearance were discarded to rule out injection artifacts.

## Calcium imaging and data analysis

5 dpf larvae in the Tg(*elavl3:GCaMP5G*) or Tg(*elavl3:GCaMP6f*) background, respectively, were individually immobilized in a drop of 2% low melting temperature agarose (Sigma-Aldrich) in a small cell culture dish. Agarose was left to dry briefly before a part was cut away and a second larvae was embedded close to the first larvae and in the same focal plane. Dish was filled with E3 medium. Calcium signals were recorded using an Olympus BX51 WI epifluorescence microscope with a 5 x objective and by means of a camera (1.33 Hz sampling rate) for 1 hr with the VisiView software established by the Visitron Systems GmbH. In each experiment, the GCaMP fluorescence signal of manually selected regions of interest (ROIs, whole brain) was extracted using Fiji ImageJ (RRID:SCR_003070) (*Schindelin et al., 2012*). Images were corrected for sample drift with Correct_3D_Drift.py script (*Parslow et al., 2014*) in Fiji ImageJ (*Schindelin et al., 2012*). For each time point, the mean intensity of each ROI was measured and further processed using a custom script in R with the RStudio interface *R Development Core Team, 2022*; *Posit team, 2023*. The baseline ($F_0$) as 1% of the total fluorescence of every ROI was computed within a moving window of 150 s (for normal brain activity) or 300 s (for *eaat2a*$^{-/-}$ seizure analysis) per fish. Subsequently, the fractional change in fluorescence ($\Delta F/F_0$) of each ROI

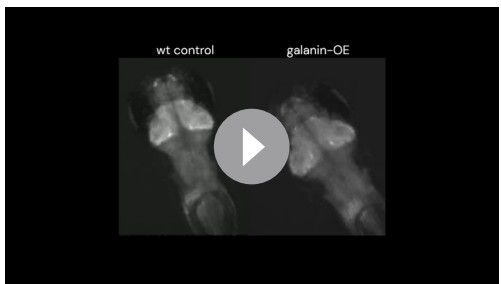

**Video 1.** Whole brain activity recording in control and *hsp70:gal* larvae. Representative raw data of recorded calcium signals (*elavl3:GCaMP5G*) recorded across the brain of 5 dpf control (left) and *hsp70:gal* (right, galanin-OE) over a 15 min time period chosen at random. The recordings were exported with 7fps. Both larvae were recorded on the same day.
https://elifesciences.org/articles/98634/figures#video1

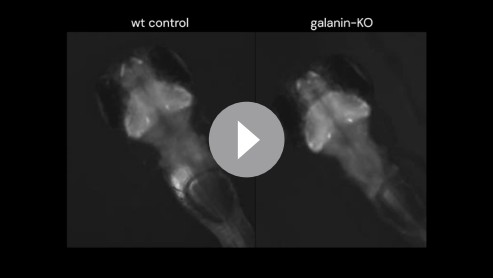

**Video 2.** Whole brain activity recording in control and *gal⁻ᐟ⁻* larvae. Representative raw data of recorded calcium signals (*elavl3:GCaMP5G*) recorded across the brain of 5 dpf control (left) and *gal⁻ᐟ⁻* (right, galanin-KO) over a 15 min time period chosen at random. The recordings were exported with 7fps. Both larvae were recorded on the same day.
https://elifesciences.org/articles/98634/figures#video2

was determined. In all imaging experiments, heartthetbeat of larvae was assessed before and after each experiment, and animals without heartbeat were excluded from analysis. In order to compare whole brain activity, thresholds for event detection were set. For examining normal brain activity, events were categorized into two distinct groups, 5% and 10% of $\Delta F/F_0$. Duration, amplitude, and number of events were calculated for each of these groups. For analyzing normal behavior, AUC was calculated for the whole normalized fluorescence trace except for the experiment with *eaat2a* mutants and the respective controls in *Figure 1B*, where AUC was calculated and averaged over two 5 min time windows per animal (same random windows for all larvae, adjusted if during seizure). To demonstrate previously published *eaat2a* hypoactivity *Hotz et al., 2022* (*Figure 1A, B, D, E and F*), data from *eaat2a⁻ᐟ⁻* (*Figure 3A-F*) and closely related wild-type larvae (*Figure 2A-F*) were re-used and differently analyzed. For this particular experiment, we removed seizures by excluding all events with amplitudes bigger than 50% $\Delta F/F_0$ and a duration of more than 20 s. For all other recordings, seizures were defined as calcium fluctuations reaching at least 100% of $\Delta F/F_0$ in the whole brain. Seizure duration was defined as the period from the first time point where $\Delta F/F_0$ is greater than 50% (seizure initiation) until the first time point where $\Delta F/F_0$ is below 50% in the brain. To generate the overlay plots (*Figures 3A, G, 4A, G and M*); all seizures were aligned by the time point where 50% of the maximum $\Delta F/F_0$ was reached. The resulting plots showcase the mean average seizure per genotype, shaded

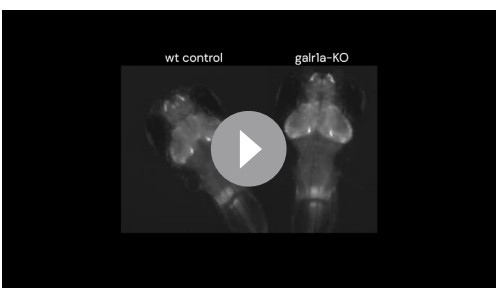

**Video 3.** Whole brain activity recording in control and galr1aKO larvae. Representative raw data of recorded calcium signals (*elavl3:GCaMP5G*) recorded across the brain of 5 dpf control (left) and galr1aKO (right) over a 15 min time period chosen at random. The recordings were exported with 7fps. Both larvae were recorded on the same day.
https://elifesciences.org/articles/98634/figures#video3

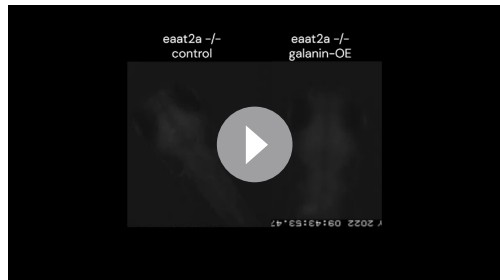

**Video 4.** Seizure recordings in control and *hsp70:gal* larvae in *eaat2a⁻ᐟ⁻* background. Representative raw data of recorded calcium signals (*elavl3:GCaMP5G*) during a typical seizure recorded across the brain of 5 dpf control (left) and *hsp70:gal* (right, galanin-OE) in *eaat2a⁻ᐟ⁻* background. The seizures were captured over 150 frames (about 1 min 50 s recording) and exported with 7fps. Both larvae were recorded on the same day.
https://elifesciences.org/articles/98634/figures#video4

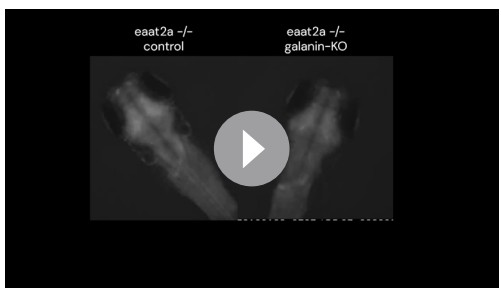

**Video 5.** Seizure recordings in control and *gal⁻/⁻* larvae in *eaat2a ⁻/⁻* background. Representative raw data of recorded calcium signals (*elavl3:GCaMP5G*) during a typical seizure recorded across the brain of 5 dpf control (left) and *gal⁻/⁻* (right, galanin-KO) in *eaat2a ⁻/⁻* background. The seizures were captured over 150 frames (about 1 min 50 s recording) and exported with 7fps. Both larvae were recorded on the same day.
https://elifesciences.org/articles/98634/figures#video5

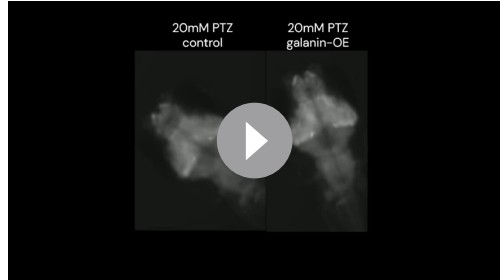

**Video 6.** Seizure recordings in control and *hsp70:gal* larvae acutely exposed to 20 mM PTZ. Representative raw data of recorded calcium signals (*elavl3:GCaMP5G*) during a typical seizure recorded across the brain of 5 dpf control (left) and *hsp70:gal* (right, galanin-OE) acutely exposed to 20 mM PTZ. The seizures were captured over 125 frames (about 1 min 50 s recording) and exported with 7fps. Both larvae were recorded on the same day.
https://elifesciences.org/articles/98634/figures#video6

area represents the standard error of the mean (SEM). AUC and time to peak of each seizure were calculated from the beginning of the aligned seizure trace. Example Videos (*Videos 1–8*) are deposited in the supplements.

## PTZ exposure

For exposure with pentylenetetrazole (PTZ), 20 mM PTZ was prepared freshly on each experimental day (Sigma-Aldrich). For acute PTZ exposure, E3 medium was replaced by 20 mM PTZ (in E3) directly before starting the 1 hr imaging period. For PTZ rebound experiments, freely swimming larvae were exposed to 20 mM PTZ (in E3) in a 50 ml cell culture dish for 1 hr, subsequently rinsed in a dish containing E3 medium, and were then transferred to a fresh dish with E3 medium for 1 hr. Larvae were embedded as described earlier and left in E3 medium until the 1 hr calcium imaging experiment was conducted precisely 2 hr after initiating the PTZ washout.

## Quantification and statistical analysis

Statistical analysis was performed using R software version 4.2.1 with the RStudio version RStudio 2023.09.1+494 interface (*R Development Core*

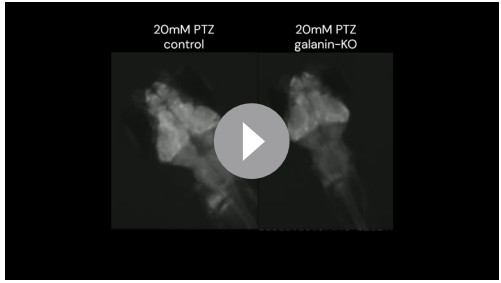

**Video 7.** Seizure recordings in control and *gal⁻/⁻* larvae acutely exposed to 20 mM PTZ. Representative raw data of recorded calcium signals (*elavl3:GCaMP5G*) during a typical seizure recorded across the brain of 5 dpf control (left) and *gal⁻/⁻* (right, galanin-KO) acutely exposed to 20 mM PTZ. The seizures were captured over 125 frames (about 1 min 50 s recording) and exported with 7fps. Both larvae were recorded on the same day.
https://elifesciences.org/articles/98634/figures#video7

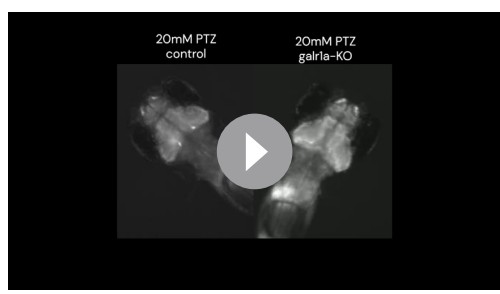

**Video 8.** Seizure recordings in control and galr1aKO larvae acutely exposed to 20 mM PTZ. Representative raw data of recorded calcium signals (*elavl3:GCaMP5G*) during a typical seizure recorded across the brain of 5 dpf control (left) and galr1aKO (right) acutely exposed to 20 mM PTZ. The seizures were captured over 125 frames (about 1 min 50 s recording) and exported with 7fps. Both larvae were recorded on the same day.
https://elifesciences.org/articles/98634/figures#video8

Team, 2022; Posit team, 2023) or GraphPad Prism version 10.1.2 for Windows (RRID:SCR_002798), GraphPad Software, Boston, Massachusetts USA, https://www.graphpad.com/. Calcium imaging data was tested for normality using the Shapiro–Wilk test. Wilcoxon-Mann-Whitney test nonpaired analysis was used for calcium imaging analysis (amplitudes, duration, AUC, time to peak). For comparing event numbers, as data was overdispersed, a negative binomial generalized linear model was fitted and assessed (package: MASS, glm.nb in R). For qPCR data, Student's t-test in GraphPad Prism was used. Differences were considered statistically significant if $p < 0.05$. Description of the number of animals used for each experiment can be found in the figures and figure legends.

## Whole-mount immunostaining

5 dpf larvae were fixed in 4% PFA, washed in PBS, and permeabilized using acetone at –20 °C. Larvae were blocked using PBDT (PBS +1% BSA+0.5% Triton X-100 +1% DMSO) with 10% goat serum for 30 min at room temperature and subsequently incubated with primary antibodies overnight at 4 °C. Rabbit anti-galanin (1:400, EDM Milipore AB5909), Mouse anti-acetylated tubulin (IgG2b, 1:500, Sigma 7451) were used as primary antibodies. Following PBDT washes, embryos were incubated in secondary antibodies Goat anti-Rabbit Alexa Fluor 488 (Invitrogen A-11008, Thermo Fisher Scientific) and goat anti-mouse IgG2b Alexa Fluor 647 (Invitrogen A-21242, Thermo Fisher Scientific) for at least 2 hr at room temperature. Larvae were cleared in an increasing glycerol row and stored in 70% glycerol in PBS. Larval heads were cut and mounted using Mowiol (Polysciences). Confocal images were acquired using the Leica TCS LSI microscope. Maximum z-projections were created using Fiji ImageJ (Schindelin et al., 2012).

## Acknowledgements

We thank P Podlasz (University of Warmia and Mazury, Poland) and U Irion (Max Planck Institute, Tübingen) for transgenic lines. We thank M Walther, K Dannenhauer, and H Möckel for excellent technical and animal support and the Neuhauss and Bachmann lab members for valuable discussions. This work was funded by the Swiss National Science Foundation (SNF 31,003 A_135598, 310030_204648) to NNR, MR, ALH, and SCFN and UZH Forschungskredit Candoc Grant K-74417-04-01 to NNR.

## Additional information

### Funding

| Funder | Grant reference number | Author |
|---|---|---|
| Schweizerischer Nationalfonds zur Förderung der Wissenschaftlichen Forschung | 310030_204648 | Stephan CF Neuhauss |
| Schweizerischer Nationalfonds zur Förderung der Wissenschaftlichen Forschung | 310030_135598 | Stephan CF Neuhauss |
| Forschungskredit Candoc | K-74417-04-01 | Nicolas N Rieser |

The funders had no role in study design, data collection and interpretation, or the decision to submit the work for publication.

### Author contributions

Nicolas N Rieser, Conceptualization, Data curation, Formal analysis, Investigation, Methodology, Writing - original draft; Milena Ronchetti, Formal analysis, Validation, Investigation; Adriana Lea Lea Hotz, Conceptualization, Investigation, Methodology; Stephan CF Neuhauss, Conceptualization, Supervision, Funding acquisition, Project administration, Writing – review and editing

## Author ORCIDs
Stephan CF Neuhauss https://orcid.org/0000-0002-9615-480X

## Ethics
Although the reported experiments do not need ethical approval, husbandry conditions were approved by local authorities (Veterinäramt Zürich TV4206).

Reviewer #1 (Public review): https://doi.org/10.7554/eLife.98634.4.sa1
Reviewer #2 (Public review): https://doi.org/10.7554/eLife.98634.4.sa2
Reviewer #3 (Public review): https://doi.org/10.7554/eLife.98634.4.sa3
Author response https://doi.org/10.7554/eLife.98634.4.sa4

---

# Additional files

### Supplementary files
MDAR checklist

### Data availability
Raw data are deposited to Dryad (https://doi.org/10.5061/dryad.m905qfvd9).

The following dataset was generated:

| Author(s) | Year | Dataset title | Dataset URL | Database and Identifier |
| --- | --- | --- | --- | --- |
| Ronchetti R, Neuhauss H | 2025 | Guardian of excitability: Multifaceted role of galanin in whole brain excitability | https://doi.org/10.5061/dryad.m905qfvd9 | Dryad Digital Repository, 10.5061/dryad.m905qfvd9 |

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
