## [Editor Report · eLife Assessment]

The study investigated the effects of the peptide galanin on brain Ca2+ activity in zebrafish, which provides a **useful** model organism for whole-brain imaging because of its transparency. They found that galanin has distinct effects on hyperactivity and expression of galanin changes after activity increases. The strength of evidence was **incomplete** particularly for some of the conclusions regarding the use of convulsants and relevance to epilepsy because of limitations to the methods and interpretations of results.

---

## [Referee Report · Reviewer #1 (Public review)]

Summary:

In this study, authors explored how galanin affects whole-brain activity in larval zebrafish using wide-field Ca2+ imaging, genetic modifications, and drugs that increase brain activity. The authors conclude that galanin has a sedative effect on the brain under normal conditions and during seizures, mainly through the galanin receptor 1a (galr1a). However, acute "stressors(?)" like pentylenetetrazole (PTZ) reduce galanin's effects, leading to increased brain activity and more seizures. Authors claim that galanin can reduce seizure severity while increasing seizure occurrence, speculated to occur through different receptor subtypes. This study confirms galanin's complex role in brain activity, supporting its potential impact on epilepsy.

Strengths:

The overall strength of the study lies primarily in its methodological approach using whole-brain Calcium imaging facilitated by the transparency of zebrafish larvae. Additionally, the use of transgenic zebrafish models is an advantage, as it enables genetic manipulations to investigate specific aspects of galanin signaling. This combination of advanced imaging and genetic tools allows for addressing galanin's role in regulating brain activity.

Weaknesses:

We have carefully reviewed the revised manuscript and the authors' responses. While the authors have attempted to address the points raised, I find that the revisions and rebuttals are insufficient and not entirely adequate. The authors seem not to have modified the manuscript in any way to take our comments into account.

In particular, many of the methodological and conceptual issues I initially raised remain unresolved. For example, the fundamental concern regarding the use of whole-brain calcium imaging - a method that may not effectively capture the localized and network-specific nature of seizure initiation and propagation - has not been adequately addressed. The authors acknowledge some limitations but do not sufficiently discuss how these affect the interpretation of their findings or propose mitigations. This could be added to the discussion section.

Additionally, the characterization of PTZ as a "stressor" remains problematic. Although the authors have retained this terminology, PTZ is widely understood to act primarily as a proconvulsant agent rather than a general stressor, and framing it otherwise continues to appear like a model-fitting rather than evidence-driven decision. The authors should consider changing the terminology throughout the manuscript and address these concerns when discussing their choice of PTZ as "stressor".

The discussion of the EAAT2 mutant model also remains incomplete. Although the authors mention preliminary transcriptome analyses, no new data were included, and it is stated that the evaluation is ongoing. Without thorough gene expression data, alternative explanations for the hypoactivity phenotype (such as changes in AMPA receptor or other critical neurotransmission-related genes) remain plausible and unaddressed. Moreover, the authors' acknowledgement that galanin upregulation is "at best one of a suite of regulatory mechanisms" further diminishes the centrality of their conclusions without sufficiently reworking the narrative of the study.

Finally, the finding that double knockout animals for EAAT2 and galanin showed little difference in seizure susceptibility compared to EAAT2 knockouts alone suggests that galanin upregulation may not play a dominant functional role, yet this important implication is not adequately reflected in the interpretation of the results.

Conclusion:

In summary, although the authors have made some efforts to respond to the critiques, I do not believe the manuscript has been substantially improved in response in R2, and I do not see reason to change my original assessment made after R1. The major conceptual and methodological concerns remain largely unaddressed, limiting the impact and validity of the study's conclusions. These concerns should be addressed not only in the rebuttal letter but also in the manuscript.

---

## [Referee Report · Reviewer #2 (Public review)]

This revised paper describes an investigation of galanin and galanin receptor signaling on whole-brain activity in the context of recurrent seizure activity or under homeostatic basal conditions. The authors primarily use calcium imaging to observe whole-brain neuronal activity accompanied by galanin qPCR to determine how manipulations of galanin or the galr1a receptor affect the activity of the whole-brain under non-ictal conditions or when seizure activity occurs. The authors use their eaat2a-/- model (introduced in their Glia 2022 paper, PMID 34716961) that shows recurrent seizure activity as well as suppression of neuronal activity and locomotion interictally. It is compared to the well-known pentylenetetrazole (PTZ) pharmacological model of seizures in zebrafish. Given the literature cited in their Introduction, the authors hypothesize that galanin will exert a net inhibitory effect on brain activity in models of seizures/epilepsy. They were surprised to find that this hypothesis was only moderately supported in their eaat2a-/- model. In contrast, after PTZ, fish with galanin overexpression showed increased seizure number and reduced duration while fish with galanin KO showed reduced seizure number and increased duration.

Previous concerns about sex or developmental biological variables were addressed, as their model's seizure phenotype emerges rapidly and long prior to the establishment of zebrafish sexual maturity. However, it remains unclear whether all seizures detected via calcium imaging alone are also seizures that are detectable at the level of animal behavior. To confirm this, a validation of the threshold used for calcium imaging of "neuronal seizures" would be required to determine if this threshold detects only "neuronal seizures" that co-occur with behavioral seizures. Overall, this study is important and convincing, and carries clear value for understanding the multifaceted functions that neuronal galanin can perform under homeostatic and disease conditions.

Additional Concerns:

- The authors have validated their ability to measure behavioral seizures quantitatively in their 2022 Glia paper but the information provided on defining behavioral seizures as they map onto seizures detected via imaging alone was limited. The definition of behavioral seizure activity as it relates to calcium fluctuations is not expanded upon in this paper, but could provide detail about how the behavioral seizures relate to a seizure detected via calcium imaging alone.

---

## [Referee Report · Reviewer #3 (Public review)]

Summary:

The neuropeptide galanin is primarily expressed in the hypothalamus and has been shown to play critical roles in homeostatic functions such as arousal, sleep, stress, and brain disorders such as epilepsy. Previous work in rodents using galanin analogs and receptor-specific knockout have provided convincing evidence for anti-convulsant effects of galanin.

In the present study, the authors sought to determine the relationship between galanin expression and whole-brain activity. The authors took advantage of the transparent nature of larval zebrafish to perform whole-brain neural activity measurements via widefield calcium imaging. Two models of seizures were used (eaat2a-/- and pentylenetetrazol; PTZ). In the eaat2a-/- model, spontaneous seizures occur and the authors found that galanin transcript levels were significantly increased and associated with reduced frequency of calcium events. Similarly, two hours after PTZ galanin transcript levels roughly doubled and the frequency and amplitude of calcium events were reduced, while the duration increased.

The authors also used a heat shock protein line (hsp70I:gal) where galanin transcripts levels are induced by activation of heat shock protein, but this line also shows higher basal transcript levels of galanin. Due to problems with whole-brain activity in wild-type larvae, the authors used the line without heat shock. They found higher level of galanin in hsp70I:gal larval zebrafish resulted in a reduction in the number of calcium events and amplitude. In contrast, galanin knockout (gal-/-) significantly increased calcium activity, indicated by an increased number of calcium events, but a reduction in amplitude and duration. Antibody staining confirmed the absence of galanin expression in gal-/- knockouts. Knockout of the galanin receptor subtype galr1a via crispants also increased the frequency of calcium events without influencing amplitude or duration.

In subsequent experiments in eaat2a-/- mutants were crossed with hsp70I:gal or gal-/- to modify galanin expression. These experiments showed modest effects, with eaat2a-/- x gal-/- knockouts showing an increased normalized area under the curve and seizure amplitude.

Lastly, the authors attempted to study the relationship between galanin and brain activity during a PTZ challenge. The hsp70I:gal larva showed increased number of seizures and reduced seizure duration during PTZ. In contrast, gal-/- mutants showed increased normalized area under the curve and a stark reduction in number of detected seizures, a reduction in seizure amplitude, but an increase in seizure duration. The authors then ruled out the role galanin a1 receptor in modulating this effect during PTZ, since the number of seizures was unaffected, whereas the amplitude and duration of seizures was increased in galr1a knockouts.

Strengths:

(1) The gain- and loss-of function galanin manipulations provided convincing evidence that galanin influences brain activity (via calcium imaging) during interictal and/or seizure-free periods. The relationship between galanin transcript levels and brain activity in figures 1 & 2 was convincing. Antibody staining also supports the absence of galanin in gal-/- mutants. Moreover, galanin transcript levels were unchanged in galr1ako brains, suggesting the lack of compensatory effects.

(2) The authors use two models of epilepsy (eaat2a-/- and PTZ).

(3) Supplementary video files for calcium imaging support the observations.

Weaknesses:

(1) I disagree with the idea that PTZ is a 'stressor'. This was raised in previous reviews and has not been acknowledged sufficiently.

(2) Although the relationship between galanin and brain activity during interictal or seizure-free periods was clear, the mechanisms that influence excitability during PTZ remain unclear. The authors show that galr1a does not mediate this effect, since seizure amplitude and duration were more severe in galr1a KO. Therefore, it remains unclear which galanin receptor is modulating this inhibitory effect.

(3) The manuscript is heavily reliant on calcium imaging for interpretation.

Additional methods could strengthen the data, translational relevance, and interpretation (e.g., acute pharmacology using selective galanin agonists or antagonists, brain or cell recordings, biochemistry, etc).

---

## [Author Response]

The following is the authors’ response to the previous reviews

**Review 1:**
Weaknesses:The weaknesses of the study also stem from the methodological approach, particularly the use of whole-brain Calcium imaging as a measure of brain activity. While epilepsy and seizures involve network interactions, they typically do not originate across the entire brain simultaneously. Seizures often begin in specific regions or even within specific populations of neurons within those regions. Therefore, a whole-brain approach, especially with Calcium imaging with inherited limitations, may not fully capture the localized nature of seizure initiation and propagation, potentially limiting the understanding of Galanin's role in epilepsy.

We agree with the reviewers that the whole brain imaging approach is both a strength and a weakness. This manuscript and our previously published paper (Hotz et al., 2022) show indeed that the seizures have a initiation point and spread throughout the brain, interestingly affecting the telencephalon last. Localized seizure initiation was not the scope of this manuscript, however also here we would have to rely on imaging techniques. Using cell type specific drivers for specific neuronal subpopulation are an interesting approach, but outside of the scope of this study. An interesting approach would also include a more detailed analysis of glia in the context of epilepsy.

Furthermore, Galanin's effects may vary across different brain areas, likely influenced by the predominant receptor types expressed in those regions. Additionally, the use of PTZ as a "stressor" is questionable since PTZ induces seizures rather than conventional stress. Referring to seizures induced by PTZ as "stress" might be a misinterpretation intended to fit the proposed model of stress regulation by receptors other than Galanin receptor 1 (GalR1).

We also agree, that a more regional approach, after having more reliable information on the expression domains of the different galanin receptors, including more information on their respective role, is an important future research direction.

The description of the EAAT2 mutants is missing crucial details. EAAT2 plays a significant role in the uptake of glutamate from the synaptic cleft, thereby regulating excitatory neurotransmission and preventing excitotoxicity. Authors suggest that in EAAT2 knockout (KO) mice galanin expression is upregulated 15-fold compared to wild-type (WT) mice, which could be interpreted as galanin playing a role in the hypoactivity observed in these animals.However, the study does not explore the misregulation of other genes that could be contributing to the observed phenotype. For instance, if AMPA receptors are significantly downregulated, or if there are alterations in other genes critical for brain activity, these changes could be more important than the upregulation of galanin. The lack of wider gene expression analysis leaves open the possibility that the observed hypoactivity could be due to factors other than, or in addition to, galanin upregulation.

We are in the process of preparing a manuscript describing a more detailed gene expression study of this and a chemically induced seizure model. Surprisingly we did not observe strong effects on glutamate receptor related genes. This does not preclude and indeed we deem it likely that additional factors play a role, e.g. other neuropeptides.

Moreover, the observation that in double KO mice for both EAAT2 and galanin there was little difference in seizure susceptibility compared to EAAT2 KO mice alone further supports the idea that galanin upregulation might not be the reason to the observed phenotype. This indicates that other regulatory mechanisms or gene expressions might be playing a more pivotal role in the manifestation of hypoactivity in EAAT2 mutants.

Yes, we agree that galanin is likely not the only player. This warrants further investigations.

These methodological shortcomings and conceptual inconsistencies undermine the perceived strengths of the study, and hinders understanding of Galanin's role in epilepsy and stress regulation.
**Review 2:**
Previous concerns about sex or developmental biological variables were addressed, as their model's seizure phenotype emerges rapidly and long prior to the establishment of zebrafish sexual maturity. However, in the course of re-review, some additional concerns (below) were detected that, if addressed, could further improve the manuscript. These concerns relate to how seizures were defined from the measurement of fluorescent calcium imaging data. Overall, this study is important and convincing, and carries clear value for understanding the multifaceted functions that neuronal galanin can perform under homeostatic and disease conditions.

We are pleased that we could dispel the initial concerns.

Additional Concerns:- The authors have validated their ability to measure behavioral seizures quantitatively in their 2022 Glia paper but the information provided on defining behavioral seizures was limited. The definition of behavioral seizure activity is not expanded upon in this paper, but could provide detail about how the behavioral seizures relate to a seizure detected via calcium imaging.

In this paper we indeed do not address behavioral seizures but focus completely on neuronal seizures as defined in the material and methods section (“seizures were defined as calcium fluctuations reaching at least 100% of ΔF/F0 in the whole brain.”). Epileptic seizures in zebrafish, either evoked by pharmacological means or the result of genetic mutations, evoke stereotyped locomotor behavior in zebrafish as described in multiple publications (e.g. Baraban et al., 2005, Berghmans et al., 2007, Baxendale et al., 2012 and references therein).

- Related to the previous point, for the calcium imaging, the difference between an increase in fluorescence that the authors think reflects increased neuronal activity and the fluorescence that corresponds to seizures is not very clear. This detail is necessary because exactly when the term "seizure" describes a degree of increased activity can be difficult to distinguish objectively.

In our material and methods section, we describe our working definition of a seizure. Seizures are easily distinguished from increased activity by being synchronized.

- The supplementary movies that were added were very useful, but raised some questions. For example, what brain regions were pulsating? What areas seemed to constantly exhibit strong fluorescence and was this an artifact? It seemed that sometimes there was background fluorescence in the body. Perhaps an anatomical diagram could be provided for the readers. In addition, there were some movies with much greater fluorescence changes - are these the seizures? These are some reasons for our request for clarified definitions of the term "seizure".

The ”pulsating” (or “flickering”) brain activity is spontaneous neuronal activity. Some areas may appear to be more active, probably by a denser packing of neurons and intrinsically more spontaneous neuronal activity. However, since we only use normalized data, this does not affect our measurements.

- While it is not critical to change, I will again note the possible confusion that the use of the word "sedative" in this context may cause. However, I do understand this is a stylistic choice.- Supplementary Figure 1B: the N values along the x-axis appear to have been duplicated and the duplications are offset and overlapping with one another by mistake.

Thank you for pointing this out. We have corrected the figure accordingly.

**Review 3:**
(1) Although the relationship between galanin and brain activity during interictal or seizure-free periods was clear, the revised manuscript still lacks mechanistic insight in the role of galanin during seizure-like activity induced by PTZ.

We agree that the mechanistic role of galanin still needs to be defined. The role is more complex that we expected, mainly due to its negative feedback properties. A complete mechanistic understanding will require a number of additional studies and is unfortunately outside of the scope of this manuscript.

(2) The revised manuscript continues to heavily rely on calcium imaging of different mutant lines. Confirmation of knockouts has been provided with immunostaining in a new supplementary figure. Additional methods could strengthen the data, translational relevance, and interpretation (e.g., acute pharmacology using galanin agonists or antagonists, brain or cell recordings, biochemistry, etc).

Cell recordings and biochemistry is challenging in the small larval zebrafish brain. We deem the genetic manipulations that we describe to be more informative than pharmacological experiments due to specificity issues.